# Optimizing Loco Regional Management of Oligometastatic Colorectal Cancer: Technical Aspects and Biomarkers, Two Sides of the Same Coin

**DOI:** 10.3390/cancers13112617

**Published:** 2021-05-26

**Authors:** Giovanni Mauri, Lorenzo Monfardini, Andrea Garnero, Maria Giulia Zampino, Franco Orsi, Paolo Della Vigna, Guido Bonomo, Gianluca Maria Varano, Marco Busso, Carlo Gazzera, Paolo Fonio, Andrea Veltri, Marco Calandri

**Affiliations:** 1Divisione di Radiologia Interventistica, Istituto Europeo di Oncologia, IRCCS, 20141 Milan, Italy; giovanni.mauri@ieo.it (G.M.); franco.orsi@ieo.it (F.O.); paolo.dellavigna@ieo.it (P.D.V.); guido.bonomo@ieo.it (G.B.); gianluca.varano@ieo.it (G.M.V.); 2Dipartimento di Oncologia ed Emato-Oncologia, Università degli Studi di Milano, 20122 Milan, Italy; 3Divisione di Radiologia, Fondazione Poliambulanza, 25124 Brescia, Italy; 4Radiodiagnostica 1 U. A.O.U., San Luigi Gonzaga di Orbassano, Regione Gonzole 10, 10043 Orbassano, Torino, Italy; andrea.garnero@unito.it (A.G.); bussomarco@gmail.com (M.B.); andrea.veltri@unito.it (A.V.); marco.calandri@unito.it (M.C.); 5Department of Surgical Sciences, University of Turin, 10124 Torino, Italy; paolo.fonio@unito.it; 6Divisione di Oncologia Medica Gastrointestinale e Tumori Neuroendocrini, Istituto Europeo di Oncologia, IRCCS, 20141 Milan, Italy; maria.zampino@ieo.it; 7Radiodiagnostica 1 U, A.O.U. Città della Scienza e della Salute, 10126 Torino, Italy; cgazzera@cittadellasalute.to.it; 8Department of Oncology, University of Turin, 10124 Torino, Italy

**Keywords:** colorectal cancer, oligometastatic disease, surgery, loco regional treatments, chemotherapy

## Abstract

**Simple Summary:**

The treatments for patients with oligometastatic colorectal carcinoma are rapidly evolving. The present review focuses on the role of minimally invasive techniques since they can now be used as an alternative to surgical management in selected cases in association with systemic therapies according to ESMO and NCCN guidelines. In recent years, biomarkers (both at molecular and imaging level) have emerged as a relevant and potential criteria for treatment strategy decision and will be crucial in the future for patients selection. Tumor molecular profile impacts on local outcome of image guide ablation as well as metabolic imaging which predicts the outcome of both percutaneous and trans-arterial treatments. Oncologists should be aware of advantages and disadvantages of those treatments options as well as the potential role of molecular profile for a better patient selection.

**Abstract:**

Colorectal cancer (CRC) is the third most common cancer worldwide and has a high rate of metastatic disease which is the main cause of CRC-related death. Oligometastatic disease is a clinical condition recently included in ESMO guidelines that can benefit from a more aggressive locoregional approach. This review focuses the attention on colorectal liver metastases (CRLM) and highlights recommendations and therapeutic locoregional strategies drawn from the current literature and consensus conferences. The different percutaneous therapies (radiofrequency ablation, microwave ablation, irreversible electroporation) as well as trans-arterial approaches (chemoembolization and radioembolization) are discussed. Ablation margins, the choice of the imaging guidance as well as characteristics of the different ablation techniques and other technical aspects are analyzed. A specific attention is then paid to the increasing role of biomarkers (in particular molecular profiling) and their role in the selection of the proper treatment for the right patient. In conclusion, in this review an up-to-date state of the art of the application of locoregional treatments on CRLM is provided, highlighting both technical aspects and the role of biomarkers, two sides of the same coin.

## 1. Introduction

Colorectal cancer (CRC) is one of the most common malignant diseases, representing the second most frequent cause of cancer-related death in USA and in Europe [1].

Approximately 50% of CRC patients will present metastatic disease during their lifetime, half of them at diagnosis; liver being the most commonly involved site [2,3].

Currently, the management of metastatic disease is a major challenge and a multidisciplinary approach is necessary to optimize results by taking into consideration clinical characteristics and molecular classifications, thereafter tailoring treatment.

In contrast to diffuse metastatic disease, the concept of oligometastatic disease (OMD) was introduced by Hellman and Weichselbaum in 1995 as a transitional state between localized and diffuse metastatic burden; in recent years, this concept has gained wide consensus in CRC management [4].

In the OMD setting, optimal local control is essential to improve outcome [5,6].

When feasible, surgery (R0 resection) seems to be the best option with the greatest likelihood of cure for patients with CRC with limited liver metastases, with 5-year survival up to 40–50% [7,8,9,10,11,12,13,14].

Unfortunately, the majority (70–80%) of patients are unsuitable candidates for resection due to clinical and/or surgical technical factors such as tumor size, location, multifocality, or inadequate hepatic reserve.

In this scenario, the role of interventional radiology is becoming increasingly important: complete ablation of all visible sites may affect local cure and may allow possible discontinuation of systemic therapy, thus inducing disease-free interval and quality of life improvement [15,16]. The European Society of Medical Oncology (ESMO) classified both surgical resection and thermal ablation as local ablative treatments (LATs) included in the treatment algorithm for OMD [6], underlining the importance of a multidisciplinary discussion when dealing with OMD patients.

For these reasons, patients with OMD should be carefully selected in order to optimize the results of the most modern technologies available even by means of specific biomarker investigation.

An extensive study is going on to biologically characterize oligometastatic CRC to provide a framework for its integrated classification and treatment [17].

The aim of our review is to report and highlight the key points of the application and the limitations of LATs performed by interventional radiologists in the setting of oligometastatic CRC treatment with special regard to the potentiality of biomarkers as predictors of LAT outcomes.

## 2. Interventional Treatment Options for OMD CRC

Interventional radiologists can provide different therapeutic strategies in the treatment of patients with oligometastatic CRC. From the technical perspective, interventional treatments can be divided in percutaneous ablative treatments and trans-arterial treatments.

### 2.1. Image Guided Ablation Techniques

Image-guided ablation techniques have been used mainly in the treatment of primary liver cancer [18], but in the last decade they have been more and more used in the treatment of oligometastatic CRC. With the application of different thermal or non-thermal energies under image guidance, it is possible to achieve the destruction of the desired amount of tissue through a percutaneously inserted applicator. Radiofrequency ablation (RFA) is by far the most often used technique for thermal ablation of the liver, given that it has been available for a longer time. More recent techniques such as microwave ablation (MWA), cryoablation, laser or irreversible electroporation (IRE) have been reported to be beneficial in this setting, showing promising results.

LATs can be performed both under general anesthesia as well as under conscious sedation, which consist of a drug-induced analgesia and depression of consciousness, during which patients respond purposefully to verbal commands, with adequate spontaneous ventilation and maintained cardiovascular function [19].

Some evidence suggest that general anesthesia should be favored over conscious sedation in order to reduce periprocedural perception of pain and increase local tumor progression-free survival [20] and expert consensus statements [21] highly recommend the diffusion of general anesthesia for the standardization of LATs. However, conscious sedation is still the most diffused approach in many western and eastern countries [22].

Table 1 shows pros and cons of the described techniques.

#### 2.1.1. Radiofrequency Ablation

Ionic friction and heat generation, associated with protein denaturation and subsequent coagulative necrosis, are the main mechanisms behind RFA [23,24,25,26,27]. A high frequency alternating current is delivered from the tip of an electrode into the surrounding target tissue [28,29]. The alternating current determines movements in the ions within the tissue resulting in frictional heating. As the temperature rises above 60 °C, cellular necrosis is seen [30]. The device usually consists of a 14–17 Gauge needle, up to 15 to 25 cm long; it may contain hook-shaped electrode arms or tines used to obtain larger and more spherically shaped ablation volumes. Even if the interventional radiologists prefer the percutaneous approach, there is no consensus as to which is the best approach of needle insertion into the tumor, whether percutaneous or laparoscopic. RFA can be performed under ultrasound (US), computed tomography (CT), or magnetic resonance imaging (MRI) guidance, as well as using fusion imaging [31] depending on lesion visibility and operator experience. To control pain and minimize patient movement the procedure is usually performed under conscious sedation or general anesthesia [16,32]. Once the needle is in the right position within the target lesion and tines have been extended or deployed into the tumor, the electrode is connected to a generator and the ablation process is performed. RFA can reproduce a defined and predictable ablation area depending on the length of the exposed tip and the presence or absence of the hook-shaped electrode arms. Tumors which measure < 3 cm in maximal diameter can be easily ablated by placing the needle electrode in the center of the tumor [33]. Tumors measuring > 3 cm may require more than one needle electrode insertion, creating an overlapping ablation zone. To effectively destroy lesions that measure ≥ 5 cm in maximal diameter with currently available RFA devices can be challenging. Furthermore, similarly to surgery [34], the achievement of at least 5 mm (ideally 10 mm) margins in all planes [34] is highly desirable in order to minimize residual disease and risk of local recurrence. Additionally, for this reason, some authors suggest a combined approach using both arterial embolization and percutaneous thermal ablation for large tumors [35].

Limitations of RFA include heat loss via nearby blood vessels (“heat-sink effect”) and injury to nearby organs caused by heat propagation [36]. RFA could be potentially dangerous when treating lesions situated in challenging locations such as the hilum; moreover, despite the coagulation effect of RFA, bleeding represents a risk in percutaneous RFA (similarly to all the other percutaneous procedures). Protective techniques such as hydro-dissection and bowel insufflation [24] can avoid such problems. Centrally located, perihilar metastases represent a poor indication for RFA; indeed their ablation could lead to biliary complications as well as lower efficacy of treatment due to surrounding large blood vessels. Nevertheless, the rate of major complications is low [37]. The adequate visualization of the target tumor seems to clearly impact the results of ablation [38]. Experimental results suggest that temperature mapping is a potential useful tool for RFA monitoring, allowing to estimate the achieved ablation zone and recognize an eventual heat sink effect [39]. Post ablation imaging (CT or MRI) is always required to assess results and monitor the ablated tissue over time. Relevantly, local efficacy of RFA has been reported by Elias et al. [40] to be equivalent to wedge resection in small metastases. The association of RFA with systemic treatment has significantly improved overall survival (OS) when compared with systemic treatment alone [16].

#### 2.1.2. Cryoablation

The main principle of cryoablation is to determine cellular damage by rapid freezing of the tissue. The cooling and subsequent thawing of the needle leads to the freezing of the surrounding tissue by convection and conduction. The early cooling effect determines the formation of intracellular ice crystals causing cell membrane damage and death. The formation of ice crystals in the capillaries feeding the tumor mass leads to ischemia [41,42]. Percutaneous cryoablation can be performed under CT, MRI or US guidance [43]. Currently, cryoablation is still considered a more expensive tool compared to RFA and other ablation techniques, thus making it less popular among interventional radiologists. Ice ball formation within the vessels or biliary ducts can lead to injury and subsequent bleeding. A rare but possible complication is cryoshock, secondary to cytokine release by necrotic tissue, resulting in a systemic syndrome characterized by fever, tachycardia, and tachypnea.

To date, data regarding the use of cryoablation for metastatic CRC are limited when compared to RFA, since it is a relatively new technique and fewer centers use cryoablation for treating liver lesions. Ng KM et al. [44] reported the results of cryoablation in 293 patients with unresectable colorectal metastases. Survival rates of 1, 3, 5 and 10 years were 87%, 41.8%, 24.2% and 13.3%, respectively. These results are less encouraging when compared to RFA results. Further studies are needed to consolidate cryoablation as a treatment option of OMD CRC.

#### 2.1.3. Microwave Ablation

In MWA, coagulative necrosis is induced by microwaves applied directly to the target tissue through a percutaneously placed antenna producing rapid temperature elevation [45,46,47]. In comparison to other ablative techniques, microwaves (between 300 MHz and 300 GHz in the electromagnetic spectrum) propagate well through all tissues including water vapor and charred desiccated tissue induced by the ablative process. As a result, microwaves provide more efficient heating than other ablation techniques, making them preferable in tissues with high blood supply or nearby vascular heat sinks [48]. Thanks to technological improvements, including applicator cooling device and power control of the microwave, this technique is very promising for the future [27,46,49,50] and is gaining popularity among interventionalists [22]. Ablation often takes less than 10 min, typically averaging 3–7 min, improving overall efficiency and reducing anesthesia time. The major complication rate of MWA was reported as 4.6% when compared to 4.1% for RFA [51]. The most common complications include hemorrhage, portal vein thrombosis, bile leak/biloma, liver abscess, pleural effusion and tumor seeding. Nowadays, MWA is probably considered the most promising technology in the interventional management OMD CRC. Indeed, many prospective registries are open (such as CIEMAR, NCT03775980) and ongoing trials (such as the COLLISION, NCT03088150, [52]) are enrolling patients in order to establish the non-inferiority of this technique compared to surgery.

#### 2.1.4. Electroporation

Electroporation is based on the use of electrical pulses created by monopolar electrodes (up to six) linked to an electrical generator, delivering a maximum of 50 A and 100 V. This creates pores in cell membranes and consequent apoptoptic cell death due to increased permeability. The magnitude of the electrical field decreases from the center outwards. Electroporation can be reversible or irreversible, the latter causing cell death [53,54]. Outlined advantages of IRE include: no “heat sink effect” and limited or no injury to the vessels and organs in close proximity to the tumor. IRE is optimal for managing tumors smaller than 5 cm [55], indeed an important limitation is the incapability to completely ablate lesions larger than 5 cm without repeated attempts or repositioning of the electrodes [56]. A recent phase II study supports the potentiality of this technology in the CLM setting [57]. Due to the limitations of RFA and MWA on hilar tumors [58], despite a non-neglectable complication rate of IRE itself in this hard-to-reach CLM 57], the most likely future potential application of this technique will be as a niche indication in this setting.

#### 2.1.5. Laser Ablation

In the last decade, lasers have been successfully used in the treatment of cancer and other diseases. Laser-induced interstitial thermotherapy (LITT) represents an effective and minimally invasive surgical technique in the treatment of various cancers such as liver, colorectal, lung, head and neck, brain, prostate and pancreas [59,60,61,62,63,64]. Near-infrared light from Nd:YAG laser or diode laser is applied given its ability to be readily and easily absorbed by human tissues [65]. Temperature distribution and dimension of the laser-induced damage are determined by thermal and optical properties of the treated tissue and by the features of the specific device [66]. One of the main advantages of LITT is the possibility to use a very small needle (21G) to reach the target lesion, thus also providing a very precise ablation area [67,68]. This makes LITT particularly valuable for lesions located in challenging positions, therefore reducing the procedure-related risks [69,70]. However, the literature is mostly focused on HCC management rather than OMD [71,72]. In the oligometastatic CRC setting, laser ablation has been applied even in the treatment of lung metastases [73]; in this context, when comparing the three different ablation methods, there were no significant differences in the time of tumor progression or in survival rates [74]. In conclusion, LITT diffusion for oligometastatic CRC treatment may be halted by the limited size of the ablation areas per single insertion, despite its potentiality for treating lesions in challenging sites.

### 2.2. Transarterial Procedures

#### 2.2.1. Transarterial Chemoembolization

In transarterial chemoembolization (TACE), cytotoxic agents are injected directly into the tumour. In general, TACE consists of the injection of different types of chemotherapic agents mixed with microspheres or embolic particles such as lipiodol oil, collagen particles, trisacryl gelatin microspheres or polyvinyl alcohol particles, which shut down the tumoral blood flow as well as stimulate the release of high doses of the drug [75,76,77,78,79]. It is proved that ischemia increases vascular permeability and thereby promotes penetration of chemotherapeutic agents into the tumor with the advantage of maximizing local cytotoxic/ischemic damage as well as minimizing systemic side effects [80,81]. Drug-eluting beads transarterial chemoembolization (DEB-TACE) is a relatively new option for the treatment of CRC metastases. Its principle is based on the intra-arterial administration of drug-loaded beads to release chemotherapeutic agent into the tumour arterial network, whilst the embolization effect limits drug washout. DEB-TACE is generally used in patients with liver only or liver-dominant metastatic disease as well as in first-line, second-line, and salvage settings, without current consensus regarding the optimal treatment option. There are limited available data on this technique from single-arm studies. Two published randomized clinical trials (RCTs) evaluated different patient cohorts: the first RCT using DEB-TACE in combination with systemic FOLFOX (5-fluorouracil + leucovorin + oxaliplatin) as first-line therapy [82], with the second comparing DEB-TACE with systemic FOLFIRI (5-fluorouracil + leucovorin + irinotecan) in the third-line setting [79]. Overall progression-free survival was not significantly different between these groups. Mauri et al. [83] demonstrated that TACE with small-size particles loaded with irinotecan (DEB-IRI) in patients with CRLM is a safe procedure and the promising results reported in terms of liver-specific progression-free survival and OS deserve further confirmation in larger prospective trials. Similarly to the hepatocellular carcinoma setting, some Authors [84,85,86] support the evidence that it is possible to obtain similar results with less toxicity using only drug-eluting polyvinyl alcohol microspheres (“beads”), without chemotherapy: in this way only, the ischaemic effect is used to treat the metastasis. However, data are still limited in the CRC setting requiring more robust evidence in the future.

#### 2.2.2. Hepatic Arterial Infusion of Chemotherapy

Hepatic arterial infusion (HAI) consists of delivering antineoplastic drugs directly into liver metastases achieving a high drug concentration in the metastatic tissue, since early-stage lesions are mainly supplied by hepatic arteries [87].

Port-catheters can be surgically or percutaneously implanted. In patients treated with surgical resection, the catheter is inserted during laparotomy. For radiological percutaneous placements, the femoral artery is the preferred approach, reaching the hepatic artery via the gastroduodenal artery. The catheter is deployed near the hepatic artery root and subsequently connected to a subcutaneous port [88]. Retrospective studies show higher efficiency and lower local complication rates in the radiological approach when compared with the surgical one [89,90]. Complications include catheter migration, arterial obstruction, catheter thrombosis and catheter rupture with an overall rate of 30% [91].

The high extraction ratios and the local drug concentrations of chemotherapeutic drugs achieved with HAI are the main features determining procedure rationale. The currently used drugs are oxaliplatin [92,93,94,95] in combination with floxuridine (FUDR) or 5-fluorouracil (5-FU). Regarding arterial infusion toxicity, Ducreux et al. demonstrated that oxaliplatin HAI has the same toxicity profile as intravenous infusion [95]. The use of intra-arterial and systemic chemotherapy combination demonstrates an improved outcome, indeed a tumor response rate up to 80% has been reported in patients treated with HAI-FUDR together with IV drugs (irinotecan/5-FU/oxaliplatin or oxaliplatin/irinotecan) [96,97].

In addition, in patients with high risk of recurrence, HAI-FUDR combined with systemic 5-FU seems to double disease-free survival when compared with treatment with systemic chemotherapy alone [98], even though this result does not imply an improved OS [99,100]. HAI can be considered a valid treatment option in patients with liver-limited disease in whom surgery or ablation techniques are not indicated. HAI can be considered as a second-line treatment if there is a poor response to first-line treatment and as salvage therapy or adjuvant therapy in unresectable liver metastases [101].

#### 2.2.3. Radioembolization

The principle of selective internal radiation therapy (SIRT), also known as transarterial radioembolization (TARE), is the selective delivery of radioactively labelled particles to the target liver lesions via the hepatic artery, since tumors receive most of their blood supply from the hepatic artery rather than from the portal vein [102,103]. Resin or glass microspheres loaded with the b-emitting isotope yttrium-90 (90Y) are injected into the hepatic artery reaching the tumor vessels that have an average radius of 2.5 mm. The tumor receives a dose higher than 120 Gy [104,105]. Healthy liver tissue involvement is limited, since it is highly sensitive to radiation (around 35 Gy) [106]. In general, radioembolization is performed as a monotherapy after systemic therapy failure [107]. Generally, SIRT procedure is well tolerated by patients, with only mild abdominal pain, fever, nausea and hepatic indices alteration occurring during the first week after treatment. Severe side effects are linked to deposition of the 90Y microspheres outside of the liver and consequent irradiation of normal tissue leading to radiation gastritis or ulceration (10%), radiation pancreatitis (<1%), and radiation cholecystitis (<1%), potentially avoided with a careful work-up [108]. Current research, however, has begun to focus on the use of SIRT in combination with chemotherapy as first-line treatment for CRC metastases [109,110,111]: more studies are required to further develop TARE and improve patients selection; in particular, research on personalised dosimetry [112] may overcome the limitations of recently published trials that do not seem to demonstrate a clear impact on OS [113].

## 3. Biomarkers

In general, despite biomarkers having gained an increasing role in the development of drugs and medical devices [114], there is still significant confusion about their definition and use in clinical practice [115].

In 2016, a joint task force of the U.S. Food and Drug Administration (FDA) and the National Institutes of Health (NIH) forged common definitions of biomarkers and made them available to the public via a continuously updated online document – the “Biomarkers, Endpoints and other Tools” (BEST) resource.

Biomarker can define a “characteristic that is measured as an indicator of normal biological processes, pathogenic processes, or biological responses to an exposure or intervention, including therapeutic interventions. Molecular, histologic, radiographic, or physiologic characteristics are types of biomarkers”. [116].

### 3.1. Molecular Biomarkers

In the era of targeted therapies, molecular biomarkers have emerged as important prognostic and predictive factors in CRC management to tailor systemic therapy and more recently also surgical and loco-regional treatments.

Tumours harboring mutations in the *RAS* family (*KRAS, NRAS, HRAS*) result in constitutive activation of the MAPK signalling pathway, and are unlikely to benefit from treatment with epidermal growth factor receptor (EGFR) antibodies [117].

*RAS* testing should be carried out on all patients at the time of diagnosis of CRC and is mandatory before treatment with the EGFR-targeted monoclonal antibodies (i.e., panitumumab and cetuximab) [6]. Either primary CRC or liver metastatic tissue can be used for *RAS* mutation testing, which have over 90% concordance in *RAS* mutational status [6,118]. Only if primary tumor or liver metastases samples are not available may other metastatic sites (lymph node or lung metastases) be used [6].

Rat sarcoma (*RAS*) viral oncogene mutations are found in up to 40% of patients with CRC and have been associated with reduced survival after resection of primary CRC and CRLMs [119].

Positive *RAS* mutation status is associated with increased positive or narrower resection margins after CRLM surgical treatment [120]. Furthermore, *RAS* mutation has been linked with higher incidence of disease recurrence and shorter OS in patients undergoing liver resection for CRLM [121].

Moreover, *RAS* mutation is associated with unsalvageable recurrence and with reduced survival after recurrence at any location after resection for CRLM [122].

Local tumor progression-free survival following percutaneous ablation of CRLMs was worse in patients with mutant *RAS* than in patients with wild-type *RAS* [123].

These findings support the hypothesis of a more infiltrative behavior of mutant *RAS* CRLM, therefore knowledge of *RAS* mutations can guide surgical and interventional radiology procedures.

Briefly, while minimal ablation margins > 10 mm should be always the procedural goal, this becomes crucial for mutant *RAS* CRLM [124]. Mutant *RAS* patients should be considered candidates for ablation only if adequate ablation margins can be planned and obtained [125].

Other important molecular biomarkers used to guide patient selection, treatment decision, risk stratification and prognostication are *BRAF* mutations, microsatellite instability (MSI) and co-occurring mutations in *RAS*/*TP53* and *APC*/*PIK3CA* [126].

*BRAF* is a protein kinase in the MAPK signalling pathway. In metastatic CRC, the most common *BRAF* mutation results in a change at residue 600 that substitutes glutamine for valine (V600E). Of patients with metastatic CRC, 5–8% carry *BRAF* V600E mutations [127]. Among these patients metastases are rarely limited to the liver, and those who undergo hepatectomy often develop disease recurrence at multiple sites, including peritoneum and lung [128,129]. Conversely, non-V600 *BRAF* mutations (harbored in 2% of metastatic CRC) correlate with significantly improved median OS compared with patients with wild-type *BRAF* and are excellent candidates for CRLM resection [130]. *RAS* and *BRAF* mutational status should be assessed simultaneously for prognostic purposes [6].

Microsatellite instability (MSI) and subsequent deficient DNA mismatch repair (dMMR) are found both in sporadic and familial CRCs. dMMR in sporadic CRC is caused by an epigenetic inactivation of *MLH1* gene, correlating with the presence of *BRAF* V600E mutations [131]. Lynch syndrome is an autosomal dominant genetic disorder associated with germline mutation in dMMR genes. Although dMMR appears to be a favourable prognostic marker, *BRAF* V600E mutations were observed in 45% of patients with deficient dMMR tumors conferring a worse prognosis [132].

The *TP53* gene encodes a transcription factor which starts cell cycle arrest, DNA repair, apoptosis and angiogenesis in response to multiple cellular [133]. Loss of function of *TP53* is present in 50–70% of patients with primary CRC [134].

Concomitant *KRAS* and *TP53* mutations promote resistance to preoperative chemo-radiation in locally advanced rectal cancer [135] and are associated with decreased OS after CRLM surgical treatment [136].

Adenomatous polyposis coli (*APC*) gene mutation lead to activation of the Wnt signalling pathway and is one of the earliest genetic events in CRC tumorigenesis; somatic *APC* mutations are observed in approximately 70% of sporadic CRCs. *PIK3CA* is a proto-oncogene encoding for a catalytic subunit of PI3K, involved in the PI3K/Akt/mTOR signalling pathway. *PIK3CA* mutations occur in 10–20% of CRC. Double mutation of *APC* and *PIK3CA* predicts poor response to preoperative chemotherapy and reduced OS in patients with CRLM resection [137]. Furthermore, preliminary data suggest that PI3K pathway mutation alone may predict longer time to local progression after CLM of SIRT [138].

Although mutational status is an important factor to consider in treatment planning, a different approach for improving patient survival is in developing biomarkers for early detection of primary and recurrent disease, at a point when traditional clinical indicators, such as radiographic signs, still are negative.

Liquid biopsy, including circulating tumor cells (CTCs), cell-free DNA (cfDNA), micro-RNA (miRNA), and exosomes, provides clinically or biologically relevant information about malignancies. Although still in development, potential applications of liquid biopsy include diagnosis, treatment response over time and minimal residual disease, in order to facilitate tailored therapy [139,140].

Despite levels of CTCs correlating with prognosis in patients with CRC [141], the clinical utility of CTC measurement is yet to be defined [6]. Conversely, cfDNA assessment is emerging as a new tool for molecular profiling with greater possible clinical implications than CTCs [6,142,143]. cfDNA seems to be more sensitive in identifying the presence of multiple clinically relevant resistance mechanisms in comparison to single-lesion tumor biopsy, due to the multiple resistance alterations in an individual patient [144].

A summary of the more important molecular biomarkers and their implication in interventional treatments is provided in Table 2.

### 3.2. Imaging Biomarkers

The need for defining and assessing surrogate imaging biomarkers in the setting of CRLM treated by image-guided therapies is a priority of the NIH and the Society of Interventional Radiology (SIR) [145].

Disease persistence and recurrence are major downsides of locoregional therapies such as ablation [146]. Consequently it is crucial to speed up the identification of patients at increased risk for ablation treatment failure by developing specific prognostic markers.

These markers can influence clinical decision-making both regarding further ablations within the same treatment session and early administration of adjuvant therapies. Snoeren et al. [147] carried pathologic evaluation of the needles after local ablation and demonstrated that adherence of proliferating tumor cells to the radiofrequency electrodes was an independent risk factor for shorter LTP-free survival. Biopsy samples from the core and the margins of the ablation zone and rapid tissue analysis using fluorescent stains could be proposed as intra-procedural indicators of complete tumor ablation [148]. Cornelis et al. [149] evaluated the potential advantage of pairing biopsy with immediately post-ablation PET/CT as on-site predictors of local tumor progression after ablation. PET/CT scan independently detects partial ablation without evidence of residual viable tumor by biopsy [149].

Likewise, it is necessary to assess imaging biomarkers of response and subsequent predictors of liver progression-free survival after radioembolization of CRLM. The Response Evaluation Criteria in Solid Tumors (RECIST) guideline, originally developed to assess response to cytotoxic chemotherapeutic agents, may not be sufficient to characterize early CRLM response treated with radioembolization in the salvage setting [150,151].

EORTC PET criteria, Choi criteria, and tumor attenuation criteria appear to be equally reliable surrogate imaging biomarkers of liver progression-free survival after radioembolization in patients with CRLM [150]. Shady et al. [152] demonstrated that PET/CT-derived metabolic volume metrics (i.e., metabolic tumor volume, MTV, and total lesion glycolysis, TLG) are significant predictors of OS post-SIRT of CRLM and seem to be more valuable than SUVmax and SUVpeak in this setting. Further investigations are required in this setting, since in other studies, diffusion-weighted MRI performed better than PET/CT in the prediction of response to therapy and OS [151] after SIRT of hepatic metastases. Furthermore, DWI can be used as a biomarker for monitoring response of CRLMs even to TACE, showing an increase in ADC values between pre- and post-treatment measurements in responding lesions [153].

## 4. Conclusions and Future Perspectives

CRC is one of the most common cancers worldwide and the treatment of OMD is challenging. Interventional radiology treatments are becoming increasingly popular both in curative and palliative management of these patients. Nevertheless, further studies are required to make these techniques fully standardized. Correct stratification and selection of patients for the right treatment at the right time during the course of the disease will be the real challenge for clinicians and researchers in the next years. Identifying and validating adequate molecular and imaging biomarkers will be the cornerstone for implementing interventional procedures in oligometastatic CRC.

## Figures and Tables

**Table 1 cancers-13-02617-t001:** Mechanism of actions and pros and cons of all different ablations techniques.

Ablation Technique	Action	Pros	Cons
RFA	High frequency alternating current determining ion friction, heat generation and coagulative necrosis	Low-price * Effective ablation of lesions < 3 cm Equivalent to wedge resection in small metastases	Difficult ablation of lesions > 5 cm Heat-sink effect ** Injury to nearby organs
MWA	Microwaves determining heat generation and coagulative necrosis	No heat-sink effect ** Larger zones of ablation Preferable for lesions > 3 cm Reduced ablation and anesthesia time	Expensive * Injury to nearby organs
Cryoablation	Ice crystal formation leading cell death and tumor ischemia	Visualization of ice-ball during procedure	Expensive * Cryoshock secondary to cytokine release
IRE	Electrical pulses determining cell membranepores and apoptosis	No heat-sink effect ** Effective for lesions < 5 cm Indication for central, perihilar lesions Limited injury to nearby vessels and organs	Expensive * Difficult ablation of lesions > 5 cm Placement of at least two applicators needed ECG-gating necessary General anesthesia necessary and operating room required
Laser Ablation	Conversion of absorbed light into heat	Smaller needle (21G) Precise ablation area Preferable for multiple small and variably sized lesions	Expensive * Limited size per single insertion of ablation areas

* If compared to other technologies; ** Heat-sink effect is heat loss via nearby blood vessels and injury to nearby organs caused by heat propagation. RFA: radiofrequency ablation; IRE: irreversible electroporation; MWA: microwave ablation.

**Table 2 cancers-13-02617-t002:** Principal molecular biomarkers with their implications in interventional treatments.

Biomarkers	% in Metastatic CRC	Action	Clinical and Prognostic Implications	Interventional Therapeutic Implications
*K-RAS*	15–50%	Constitutive activation of the MAPK signalling pathway	Higher recurrence rate after liver or lung CRC metastases ablation Reduced OS	• Larger ablation margins are strictly required for *RAS* mutant metastases
*BRAF*	1–8%	Constitutive activation of the MAPK signalling pathway	*V600E mutation* Recurrence after resection in multiple site (peritoneum and lung)	
			*Non-V600 mutation* Significantly improved OS compared with wild-type *BRAF*	• Excellent candidates for CRLM local therapies
MSI	2–3%	Deficient DNA mismatch repair		
*TP53*	50–70%	Cell proliferation and cell death dysregulation	Concomitant *TP53* and *KRAS* mutation associated with decreased OS after CRLM resection	
*APC*	42–70%	Constitutive activation of Wnt signalling pathway	Concomitant *APC* and *PIK3CA* mutation associated with poor prognosis after CRLM resection	• PI3K pathway mutation predicts longer time to local progression after radioembolization of CRLM
*PIK3CA*	6–28%	Constitutive activation of PI3K/Akt/mTOR signalling pathway		

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
