# Peer review of "Optimizing Loco Regional Management of Oligometastatic Colorectal Cancer: Technical Aspects and Biomarkers, Two Sides of the Same Coin"

_cancers, 2021, doi:10.3390/cancers13112617_

Round 1

Reviewer 1 Report

Very nice comprehensive review

Author Response

Thank you a lot for the positive comments and review.

Reviewer 2 Report

Authors present a comprehensive overview covering local regional therapies and biomarkers which are relevant in the treatment of oligometastatic CRC. This is an interesting topic for the readers of CANCERS. Overall quality of the manuscript including structure and tables is already very high. 

Nevertheless, I have some suggestions for minor corrections and additions.

Furthermore, I would recommend the revision of the manuscript by a native speaker. 

Simple Summary: ok

Abstract: ok

Introduction: ok

Page 2; line 92: Please, replace “needle” by applicator.

Page 3; table 1: Please, add “placement of at least two applicators needed” and “ECG-gating necessary” to Cons/IRE

Page 3; line 108: I suggest to add “ … to obtain large and more spherically shaped ablation volumes”

Page 4; line 133: Reference [21] refers to an article dealing with renal tumor ablation, better choose an article dedicated to liver ablations and hydrodissection like Levit E et al. Acta Radiol. 2012 or Rhim H et al. AJR Am J Roentgenol. 2008 

Page 4, line 138: Non-invasive MR-based thermometry should be mentioned as a potential tool for monitoring, e.g. Rempp H et al. Cardiovasc Interv Radiol 2012

Page 5, line 195: Regarding complications and clinical outcomes of IRE, please, refer to Distelmaier M et al. Radiology 2017

Page 6, line 223: In my opinion, the sentence “In general the injection of cytotoxic agents is combined with the use of Lipiodol: …” is misleading. TACE consists of the transarterial application of chemo + embolic agent which can be PVA-beads, degradable starch particles, gelatine, lipiodol, … So, please, rephrase this sentence.

Page 6, line 249: Same as above, this is misleading because the use of drug-eluting beads not only relies on ischemia but also on the effect of the eluted chemo agent.

Page 7, line 298: My suggestion is to mention the trend towards a personalized dosimetry to improve outcomes in the future. Garin E et al. Lancet Gastroenterol Hepatol 2021.

Page 10, line 434: There is some data available, that MRI-techniques as simple T2 or DWI are possible imaging biomarker for response assessment. Thüring et al. PLoS One 2020, Barabasch A et al. Radiology 2018, Lahrsow M, etal. Cardiovasc Interv Radiol 2017

Reviewer 3 Report

The authors reviewed various interventional techniques for the treatment of colorectal liver metastases (CRLM) with the description of molecular and imaging biomarkers. This review would be interesting and useful for readers. However, some important issues are lacking with regard to anesthesia and imaging biomarkers. 

#1 Anesthesia
Various degrees of invasiveness are inherent in local ablative techniques (LAT). Invasiveness for patients would be different between local and general anesthesia. Please consider adding some discussions from the aspect of anesthesia.

#2 Imaging biomarkers
Although molecular biomarkers would be suggestive for physician and interventional radiologists, the paragraph of "3.2. Imaging Biomarkers" provides an inadequate amount of information. A literature review should be performed about the following issues:
1) Dynamic CT/MRI study versus PET imaging
2) When should we acquire images before and after treatment? Please describe the appropriate timing of image acquisition for each modality.

Minor issues:
#Tables: Please delete “•”s and adjust the width.
#Abbreviations: Please check abbreviations. I found some abbreviations defined more than twice.

Round 2

Reviewer 3 Report

This review manuscript discussed the interventional techniques for colorectal liver metastases (CRLM) and locoregional therapeutic strategies with mutational biomarkers. The revised manuscript is much clearer for readers to understand and overviews both technical aspects and the role of biomarkers.

Minor issues:
#1 Manuscript and tables need some improvement, like spaces and dots.
#2 The abstract can be improved to reflect the manuscript itself.

Author Response

Dear Reviewer 3,

Thank you so much for the time you spent evaluating our manuscript and thank you for the positive comments.

The abstract was designed in order to carefully describe the most relevant aspects of the manuscript and it received generally positive comments from the other reviewers. We really would not be able to further improve it. However, if you let us know exactly what still is not working we would be more than happy to change it

Minor grammatical errors will be fixed during the next step of the publishing process.

Once again thank you for your expertise and help.

Dr Marco Calandri and Dr Lorenzo Monfardini